# A Pentavalent *Shigella flexneri* LPS-Based Vaccine Candidate Is Safe and Immunogenic in Animal Models

**DOI:** 10.3390/vaccines11020345

**Published:** 2023-02-03

**Authors:** Vladimir A. Ledov, Marina E. Golovina, Biana I. Alkhazova, Vyacheslav L. Lvov, Alexander L. Kovalchuk, Petr G. Aparin

**Affiliations:** 1Laboratory of Carbohydrate Vaccines, National Research Center-Institute of Immunology, Federal Medical Biological Agency of Russia, 24, Kashirskoe Shosse, 115478 Moscow, Russia; 2ATVD-TEAM Co., Ltd., 115522 Moscow, Russia; 3Laboratory of Preparative Biochemistry, National Research Center-Institute of Immunology, Federal Medical Biological Agency of Russia, 24, Kashirskoe Shosse, 115478 Moscow, Russia

**Keywords:** *Shigella flexneri*, vaccine, lipopolysaccharide, preclinical studies

## Abstract

A multivalent vaccine is much needed to achieve protection against predominant *Shigella* serotypes. Recently, we demonstrated the clinical applicability and immunogenic potential of tri-acylated *S. flexneri* 2a lipopolysaccharide (Ac_3_-S-LPS). Using a similar approach, we designed a pentavalent LPS candidate vaccine against *S. flexneri* 1b, 2a, 3a, 6, and Y (PLVF). In this study, we performed molecular and antigenic characterization of the vaccine candidate and its preclinical evaluation. There were no signs of acute toxicity after subcutaneous administration of PLVF in rabbits at a proposed human dose of 125 μg. No pyrogenic reactions and adverse effects associated with chronic toxicity after repeated administration of PLVF were revealed either. The immunization of mice with PLVF led to ≥16-fold increase in *S. flexneri* 1b-, 2a-, 3a-, 6-, and Y-specific antibodies. In a serum bactericidal antibody (SBA) assay, we registered 54%, 66%, 35%, 60%, and 60% killing of *S. flexneri* 1b, 2a, 3a, 6, and Y, respectively. In the guinea pig keratoconjunctivitis model, the efficacy was 50% to 75% against challenge with all five *S. flexneri* serotypes. These studies demonstrate that PLVF is safe, immunogenic over a wide range of doses, and provides protection against challenge with homologous *S. flexneri* strains, thus confirming the validity of pentavalent design of the combined vaccine.

## 1. Introduction

Dysentery is an acute infectious colitis caused by an enteric bacterial pathogen known as *Shigella*. Dysentery is ubiquitous. About 188 million global cases of shigellosis are reported annually [1]. The main risk groups for shigellosis are composed of children under 5 years of age, underprivileged elderly people, and travelers in endemic areas [2,3]. Shigellosis caused by *Shigella flexneri* is predominantly found in low- and middle-income countries [1]. Despite the high epidemiological significance of Flexner’s dysentery and progressive antibiotic resistance, a licensed vaccine against *S. flexneri* has not yet been developed.

*Shigella* vaccine development is being pursued in several directions: a live attenuated vaccine [4], a whole-cell inactivated vaccine [5], and a conjugated vaccine, including chemically conjugated or the most modern bioconjugate preparations (O-polysaccharide (O-PS) conjugated to the carrier protein, exotoxin A of *P. aeruginosa*, using gene engineering techniques) [6].

Live and inactivated whole-cell vaccines against *S. flexneri* have been immunogenic in clinical and preclinical studies [7,8,9]. However, a large number of observed adverse events and short-lived immune responses in clinical study subjects limits their regulatory approval and use in the clinic [10]. To reduce the adverse events of whole-cell vaccines, the oral route of administration [7] and/or attenuated strains have been used [4]. The main challenge for whole-cell vaccine developers is to strike a balance between reactogenicity and immunogenicity [11].

According to published clinical studies, conjugated dysentery vaccines, including bioconjugates, are safe. An increase in serum-specific antibodies of IgG and IgA classes by 2–10 times was recorded one month after the first vaccination, and levels remained unchanged for 56 days after the repeated vaccination [12].

*S. flexneri* can be divided into 13 serotypes: 1a, 1b, 2a, 2b, 3a, 3b, 4a, 4b, 5a, 5b, 6, X, and Y. Type antigens are determined by saccharide residues linked to the main chain. Additional sugars or conformation of the chemical structure form group antigens (serotypes X and Y lack type antigens and have only group antigens) [13]. Serotypes/serogroups isolated from approximately 90% of patients with dysentery include *S. flexneri* 1b, 2a, 2b, 3a, and 6 [14].

Thus, the morbidity of shigellosis caused by *S. flexneri*, in contrast to shigellosis caused by *S. sonnei*, is due to the presence of multiple serotypes of the pathogen. When developing a vaccine against shigellosis caused by *S. flexneri*, in addition to solving the problem of creating an effective vaccine immunogen, the choice and technological development of a set of antigenic vaccine components that provide protection against infection in a given geographic area are equally important.

Protective immunity to *Shigella* is induced by O-polysaccharide part of lipopolysaccharide (LPS), but LPS is highly endotoxic [15]. Therefore, the use of LPS in its native form (nLPS) as an active component of a vaccine is impossible. In an earlier study, we applied the LPS detoxification method to create a modified tri-acylated highly homogenous LPS form (Ac_3_-S-LPS) from *S. flexneri* 2a [16]. This paper describes a novel *Shigella* vaccine based on original pentavalent LPS preparation that, in addition to Ac_3_-S-LPS from *S. flexneri* 2a, includes four other Ac_3_-S-LPS compounds from *S. flexneri* 1b, 3a, 6, and Y (PLVF). The preclinical safety profile of PLVF was studied in rabbits. Antibody levels after immunization of mice with PLVF were assessed. Functional evaluation of anti-PLVF antibodies’ potency for lysis of virulent *Shigella* of selected serotypes was also performed. The protective potency of PLVF was assessed in a Sereni test.

## 2. Materials and Methods

### 2.1. Bacterial Strains and Growth Conditions

The biomass of strains of *S. flexneri* 1b 1818, 2a 1605, 3a 2167, 6 281-55, and Y 2643, which represented the smooth (S) form of bacteria each serotyped with a homologous *Shigella* serum, was obtained through fermentation using Hottinger’s broth (Nutrient Media, State Research Center for Applied Biotechnology and Microbiology, Obolensk Russia) in a 250 L fermenter (BioR 250, Prointech-bio, Pushchino, Russia) with stirring and forced aeration. Bacterial cells were separated from the liquid phase by flow centrifugation (Z-41, CEPA, Lahr, Germany). Wet cells were subsequently washed with sterile saline and water, and then lyophilized.

For serum bactericidal antibody (SBA) assay, *S. flexneri* 1b, 2a, 3a, 6, and Y were streaked on tryptone soy agar plates and incubated overnight at 37 °C. Then, bacterial cells were washed off plate surfaces using sterile 0.9% NaCl and further diluted in phosphate-buffered saline.

### 2.2. Isolation and Degradation of S-LPS

Freeze-dried bacterial cells were extracted with 45% aqueous phenol (Sigma-Aldrich, St. Louis, MO, USA) at 68–70 °C. The aqueous phase was separated, dialyzed, and lyophilized to give a crude LPS preparation. The preparation was dissolved in TRIS buffer containing 0.01% (*w*/*w*) CaCl_2_ and MgCl_2_ solution (pH 7.2). RNAse (Sigma-Aldrich, St. Louis, MO, USA) (100 μg mL^−1^) and DNAse (Sigma-Aldrich, St. Louis, MO, USA) (10 μg mL^−1^) were added, and the solution was stirred at 37 °C for 16 h. The reaction mixture was treated with proteinase K (Sigma-Aldrich, St. Louis, MO, USA) (20 μg mL^−1^) for 2 h at 55 °C and then dialyzed using ultrafiltration with a 50 kDa cut-off membrane (Vladisart, Vladimir, Russia), concentrated, and lyophilized to give a purified LPS preparation. S-LPS was obtained using gel-permeation chromatography on Sephadex G-150 (Sigma-Aldrich, St. Louis, MO, USA) in the presence of Na-deoxycholate (Sigma-Aldrich, St. Louis, MO, USA) as detergent. Fractions that contained S-LPS were combined and freeze-dried.

After the removal of lipid A (2% AcOH, 100 °C, 1 h), the carbohydrate portions of nLPS and S-LPS were profiled by an Agilent 1260 HPLC system with UV- and RI-detection on a TSK gel G3000PW (Toyopearl, Mainz, Germany) (7.8 mm I.D. 30 cm) column using 0.2 M NaCl elution solution and a flow rate of 0.5 mL min^−1^. The content of low molecular mass compounds in S-LPS did not exceed 5–10%.

Partial de-acetylation of S-LPS was performed when S-LPS (300 mg) solution was heated with stirring in 8.3% aqueous ammonia and water (100 mL) containing 100 mg of Na-deoxycholate at 30 °C for 8 h, then cooled to 5–10 °C, diluted with 200 mL of water, neutralized with AcOH, and freeze-dried. The product was treated with 100% ethanol (200 mL), the precipitate was separated by centrifugation, washed with 100% ethanol (2 × 200 mL), vacuum-dried, dissolved in water, and freeze-dried to give Ac_3_-S-LPS (213 mg) [17].

### 2.3. Production of PLVF

The candidate vaccine product was manufactured based on Ac_3_-S-LPS compounds from *S. flexneri* 1b, 2a, 3a, 6, and Y (PLVF) as the active substances at a dose of 125 μg (25 μg of each antigen compound) and contained the following formulation excipients: phenol (preservative) 0.75 mg, NaCl 4.15 mg, Na_2_HPO_4_ 0.052 mg, and NaH_2_PO_4_ 0.017 mg (all Sigma-Aldrich, St. Louis, MO, USA), and 0.5 mL sterile pyrogen-free water for injection [16]. The final form of PLVF product was formulated and dispensed aseptically in ampoules at a GMP-compliant manufacturing suite of vaccine enterprise of the Chumakov Institute of Poliomyelitis and Viral Encephalitides, Russian Academy of Sciences.

### 2.4. Chemical and Physical Analyses

SDS-PAGE was performed on a 12% acrylamide gel according to the Laemmli method using a Bio-Rad Mini-Protean electrophoresis system. Gels were stained with silver nitrate reagent (Sigma-Aldrich, St. Louis, MO, USA).

Electrospray ionization high-resolution mass spectra were recorded in the negative ion mode using a micrOTOF II instrument (Bruker Daltonics, Billerica, MA, USA). A sample (~50 ng μL^−1^) was dissolved in a 1:1 (*v*/*v*) water–acetonitrile mixture and sprayed at a flow rate of 3 μLmin^−1^. End plate offset voltage was set to 0.5 kV and capillary voltage to 4.5 kV. Drying gas temperature was 180 °C. Mass range was from *m*/*z* 50 to 3000 Da.

^1^H- and ^13^C-NMR spectroscopy were performed for solutions in 99.95% D_2_O at 323 K on a Bruker DRX-500 spectrometer (Bruker Daltonics, Billerica, MA, USA) using sodium3-trimethylsilylpropanoate-2,2,3,3-d_4_ (δ_H_0) and acetone (δ_C_ 31.45) as references for calibration. Prior to analysis, samples were freeze-dried from 99.5% D_2_O. The Bruker Topspin 2.1 program was employed to acquire and process the NMR data.

### 2.5. Animal Studies

The study on animal models was carried out in accordance with the ethical principles approved by the order of the Ministry of Health of the Russian Federation No. 199n from 4 January 2016. The study protocols were approved by the local ethics committees of the research organizations. Animals were housed in accredited animal facilities with free access to food and water. Before the start of a study, animals were placed in a separate room for a period of quarantine (14–21 days, depending on the animal species) and health-monitored.

### 2.6. Pyrogenicity

The standard method for determining the pyrogenicity is the rabbit pyrogen test (RPT). RPT was conducted on 21 adult Chinchilla rabbits (aged 3 months at the start of the experiment and weighing 2.8–3.05 g). Rabbits were randomized by weight into 7 groups. Three rabbits per group were intravenously injected with PLVF, each Ac_3_-S-LPS vaccine component or unmodified nLPS of *S. flexneri* 2a (as pyrogenicity control) at a dose of 0.025 µg kg^−1^.

A substance was considered apyrogenic if the cumulative temperature rise of three rabbits did not exceed 1.15 °C in accordance with the European Pharmacopoeia requirements [18]. The PLVF doze used for RPT was chosen by analogy to the WHO-approved Vi-vaccine pyrogenicity test dose of 0.025 µg kg^−1^.

### 2.7. Toxicology Study

A standard PLVF toxicity study was performed in rabbits. Chinchilla male and female rabbits were purchased from the Federal State Budgetary Institution of Science “Scientific Center for Biomedical Technologies of the Federal Medical and Biological Agency”, Russia. Acute toxicity studies were performed with a single subcutaneous injection of 125 μg/0.5 mL of PLVF (5 male and 5 female rabbits per group, aged 3 months at the start of experiment and weighing 2708 ± 71 g), followed by observation for 14 days. Animals in the control group were injected with 0.5 mL of PBS. Chronic toxicity studies were performed with daily subcutaneous immunization of rabbits with 125 μg/0.5 mL PLVF for a week (5 males and 5 females per group), followed by observation for 7 days. Animals in the control group were injected with 0.5 mL of PBS. Blood samples for hematology and biochemistry were collected before vaccination, and 7 and 14 days after vaccination. On the 14th day after vaccination, necropsy, histological examination of 16 internal organs and tissues of each rabbit, and the blood differential test were performed. The local irritation was studied by histopathological evaluation of the site of repeated subcutaneous injection of the vaccine preparation (macro- and microscopic description of the skin at the injection site). 

Tissue samples for histological examination were fixed in 10% neutral buffered formalin, dehydrated in ascending concentrations of alcohol, and embedded in paraffin. Paraffin sections 5 μm thick were cut on a SM 2000R microtome (Leica, Wetzlar, Germany), stained with hematoxylin and eosin, and examined using a DM1000 microscope (Leica, Wetzlar, Germany). 

The study of acute and chronic toxicity, as well as the study of the local tolerance of PLVF, was carried out in accordance with the recommendations and the requirements of local legislation (Federal Law of 12.04.10 N 61 “On the Circulation of Medicines”).

### 2.8. Immunochemical Identification of PLVF

We utilized standard sandwich ELISA with monovalent rabbit antisera against type antigens I, II, III, and VI, and group antigen 3,4 (Agnolla®, SPbSRIVS, Saint Petersburg, Russia) and PLVF-coated microwells for identification of PLVF components 1b, 2a, 3a, 6, and Y, respectively, and intact *S. flexneri* S-LPS 1b-, 2a-, 3a-, 6-, and Y-coated microwells for control antigenic activity.

### 2.9. Immunogenicity in Mice

Sixty (CBA × C57BL/6) F1 female mice (5 mice per group, 8-weeks-old at the start of the experiment and weighing 18 ± 0.3 g) were purchased from the Federal State Budgetary Institution of Science “Scientific Center for Biomedical Technologies of the Federal Medical and Biological Agency”, Russia and immunized intraperitoneally with 125 μg per mouse of PLVF or with two doses of 25 μg and 50 μg of Ac_3_-S-LPS 1b, 2a, 3a, 6, and Y. Two weeks after the primary injection, the mice were boosted with the same dose. At day 15 after the secondary immunization, serum samples were collected and levels of LPS-specific total IgG and IgM were evaluated by ELISA (using a standard protocol) with native *S. flexneri* LPS (nLPS) 1b, 2a, 3a, 6, and Y adsorbed on microplates (Greiner, Kremsmünster, Austria). Control animals were given saline.

### 2.10. SBA Assay

Heat-inactivated mouse serum samples were diluted 1:3000 in PBS and 100 µL were added into 96-well U-bottom plates (Medpolymer, Saint Petersburg, Russia). A 100 µL aliquot containing 10^4^ CFU of each *S. flexneri* serotype 1b, 2a, 3a, 6, and Y separately and 25 µL of guinea pig complement (made in-house) were added into the wells. The final volume was 300 µL and serum dilution 1:9000. Plates were incubated for 1.5 h at 37 °C in a shaker at 200 rpm. Serum bactericidal activity was assessed individually for each mouse. The assay included complement control wells containing *S. flexneri* bacteria with guinea pig complement with no serum. This complement control was used to define 0% killing in the SBA killing rate calculation.

The percentage of bacteria killing (SBA rate) was determined by the equation [1- (colony forming units (CFU) of surviving bacteria/total CFU)] × 100.

### 2.11. Efficacy Evaluation of PLVF

Male (Agouti strain) guinea pigs were purchased from the Federal State Budgetary Institution of Science “Scientific Center for Biomedical Technologies of the Federal Medical and Biological Agency”, Russia. To examine the PLVF for eye protection against virulent *S. flexneri* stains, groups of 10 guinea pigs (aged 3 months at the start of the experiment and weighing 275 ± 3 g) were twice immunized subcutaneously dorsally with a dose of 125 µg of vaccine at an interval of 10 days. Control animals were given saline. Ten days after the last immunization, *S. flexneri* keratoconjunctivitis was induced in experimental and control groups of animals by inoculation into the mucosal surface of the conjunctiva of both eyes with a suspension of a virulent strain at a dose of 2 × 10^9^ cells in 30 µL of sterile saline. Each group was inoculated with one of *S. flexneri* 1b, 2a, 3a, 6, and Y serotypes. Keratoconjunctivitis was assessed 7 days after challenge by visual inspection. The efficacy of PLVF was calculated by the formula: PLVF Efficacy = 100 × (Control attack rate – PLVF attack rate)/Control attack rate, where attack rate = number of infected eyes/total eyes [4].

### 2.12. Statistical Analysis

Statistical analyses of immunogenicity data were performed using GraphPad Prism 7 software (GraphPad Software Inc., La Jolla, CA, USA). For statistical significance while comparing groups of mice, the Student’s t test was performed. Statistical significance was defined as *p* < 0.05.

## 3. Results

### 3.1. Chemical Structure and Specific Epitope Identification of PLVF Compounds

PLVF contains modified long-chain S-LPS compounds with a length of O-PS chain of 15–25 repeating units (Figure 1a). Mass spectrum analysis demonstrates that lipid A of 1b, 2a, 3a, 6, and Y represents highly homogenous Ac_3_-S-LPS and contains mainly three (the highest peak at *m*/*z* = 1053) residues of unsaturated fatty acids (Figure 1b). It should be noted that no peaks for penta- and hexa-acyl derivatives were present in the mass spectra of the modified lipid A from *S. flexneri* serotypes 1b, 2a, 3a, 6, and Y.

The characteristic signals in the anomeric carbon region of the ^13^C NMR spectra of *S. flexneri* Ac_3_-S-LPS do not overlap and are a distinctive feature for each serotype, notably the characteristic rhamnose signals associated with different degrees and positions of O-acetylation. The shifts of anomeric protons were analyzed by comparing the NMR spectra of *S. flexneri* 1b, 2a, 3, Y, and 6 Ac_3_-S-LPS with the literature data (Figure 2).

For immunochemical detection of PLVF components 1b, 2a, 3a, 6, and Y, we used monovalent *Shigella* antisera against type-specific antigens I, II, III, and VI, and group-specific antigen 3,4, respectively. Each Ac_3_-S-LPS component in PLVF composition was antigenic (or serologically active) and reacted with the corresponding specific antisera, thus confirming the presence of type- or group-specific epitopes (Figure 3). The serological activity of Ac_3_-S-LPS components of the vaccine was comparable to that of nLPS of the respective serotypes.

### 3.2. Pyrogenicity 

Pyrogenicity of PLVF or PLVF’s individual components 1b, 2a, 3a, 6, and Y was examined in rabbits and compared with *S. flexneri* nLPS. PLVF (∑Δ t °C = 0.9 °C) or PLVF’s components were apyrogenic − ∑Δ t °C < 1.15 °C when injected intravenously into rabbits (Figure 4). At the same time, unmodified nLPS induced a strong pyrogenic reaction in rabbits − ∑Δt °C (nLPS) = 4.2 °C.

### 3.3. Immunogenicity in Mice

In all groups of mice immunized with Ac_3_-S-LPS 1b, 2a, 3a, 6, and Y, a significant increase in the titer of specific IgG and IgM was observed compared with the intact animals. The antibody response to five Ac_3_-S-LPS compounds did not change with an increase in immunization dose from 25 μg to 50 μg (Figure 5a,b). The immunogenicity in mice of low-endotoxic Ac_3_-S-LPS from *S. flexneri* 1b, 2a, 3a, and Y was almost equal. The level of serum antibody response in mice immunized with Ac_3_-S-LPS serotype 6 was higher.

In the group of mice immunized with PLVF, an increase in the titer of specific IgG antibodies was also observed (*p* ≤ 0.05). Titers of IgG to LPS *S. flexneri* 1b, 2a, 3a, Y, and 6 were 1213 ± 438, 1393± 980, 1056± 438, 2786 ± 2629, and 1600 ± 2360, respectively (Figure 6a).

In the groups of mice immunized with only one component of PLVF, the titer of IgG was 1393 ± 980 for 1b, 1213 ± 1403 for 2a, 1213 ± 438 for 3a, 2425 ± 2147 for 6, and 1213 ± 1043 for Y. These results were not significantly different from those in the group of mice immunized with PLVF. At the same time, titers in both experimental groups were at least 16 times higher than in the control group (Figure 6b). We did not register the occurrence of antigenic competition and suppression of the response when closely related polysaccharide antigens were combined into a single preparation.

Next, we evaluated the functional activity of *S. flexneri* 1b-, 2a-, 3a-, 6-, and Y-specific antibodies in sera from mice immunized with PLVF at a dose of 125 μg using the SBA assay. Insignificant killing (<15%) of *S. flexneri* of different serotypes was observed using non-immune sera (Figure 7). After PLVF immunization, we registered significant rises in mouse immune sera bactericidal activity against virulent *S. flexneri* strains. The estimated killing rate was 54%, 66%, 35%, 60%, and 60% for *S. flexneri* 1b, 2a, 3a, 6, and Y serotype/serogroup, respectively. These data directly demonstrate the ability of modified S-LPS from different strains of *S. flexneri* to act as an effective protective antigen.

### 3.4. PLVF Protects against Keratoconjunctivitis in a Guinea Pig Model after Separate Infection with Each of S. flexneri 1b, 2a, 3a, 6, and Y Serotypes

To protect against conjunctivitis, guinea pigs were immunized with PLVF twice subcutaneously with an interval of 10 days. On day 10 after the last immunization, groups of animals were challenged with ID_80_ of virulent strains of *S. flexneri* 1b, 2a, 3a, 6, and Y. On day seven after infection, visual assessment showed that in groups of immunized animals the efficacy was 69%, 75%, 50%, 50%, and 69% against *S. flexneri* serotypes 1b, 2a, 3a, 6, and Y, respectively (Figure 8). In unimmunized control animals, at least 80% of the eyes were infected depending on the serotype of the virulent strain of *S. flexneri*.

### 3.5. Toxicology Study

To assess the general toxicity, the effect of the vaccine preparation was studied in rabbits using single and repeated subcutaneous administration. The results of the acute toxicity study showed that PLVF was non-toxic at a dose of 125 μg. Animal mortality was not observed, there were no signs of toxicity, and there were no changes in body weight or hematological and biochemical tests. No pathological changes in bone marrow hematopoiesis were observed in all groups. There were no significant histological differences found in organs from PLVF and control group (Appendix A, Appendix A).

The results of the chronic toxicity study showed that daily subcutaneous application of PLVF to rabbits did not cause disturbances in the functional state of the main organs and systems of the body.

There was no local irritating effect in the area of PLVF administration after repeated subcutaneous application to rabbits.

## 4. Discussion

Flexner’s shigellosis represents a longstanding, difficult public health problem in many countries around the world. The achievement of field protection after vaccination against *S. flexneri* is an extremely difficult task.

*Shigella*-specific antibodies play an important role in promoting host defense against shigellosis. Children and adults living in areas endemic for shigellosis develop circulating antibody secreting cells and serum antibodies specific for *Shigella* LPS and invasion plasmid antigen. Sera of volunteers infected with *S. flexneri* 2a have a pronounced bactericidal efficacy [19].

Recent studies of functional antibodies after immunization with protein-lipopolysaccharide outer membrane vesicles (OMV) *S. sonnei* vaccine based on generalized modules for membrane antigen (GMMA) have shown that anti-LPS antibodies are the main drivers of bactericidal activity. On the contrary, anti-protein antibodies had limited ability to either bind to *Shigella* cells or kill them in the presence of complement [20].

In modern LPS-enriched *Shigella* vaccines (Invaplex_AR-Detox_, GMMA) genetically modified Shigella strains (Δ*msb*B orΔ*htr*B) are being used to prevent production of the most endotoxic form of LPS with hexa-acylated lipid A domain [21,22]. However, in these vaccines, LPS remains heterogeneous like classical endotoxin and contains penta- and tetra-acylated lipid A and, therefore, is still relatively endotoxic and pyrogenic. As a result, the doses of preparations containing genetically modified LPS proposed for the safe parenteral administration by the authors of the abovementioned studies are relatively low (in the range of 0.1–15 μg). Additional fractions of protein antigens, Ipa molecules (Invaplex_AR-Detox_) or OMV proteins (GMMA), were also present in these vaccine preparations. Further studies aiming to increase the content of LPS, and thus change the design of GMMA-based vaccines to enhance their immunogenicity and efficacy, have been announced [20].

Our previous studies have shown that Ac_3_-S-LPS is apyrogenic and is the most immunogenic form of *S. flexneri* 2a S-LPS [16]. As we have already demonstrated the possibility of safe clinical use of *S. flexneri* 2a Ac_3_-S-LPS, it was reasonable to use this approach to obtain Ac_3_-S-LPS components of other serotypes/serogroups for the development of PLVF. Therefore, PLVF was designed to include in the vaccine dose the maximum number of *S. flexneri* antigens. Thus, PLVF contains five Ac_3_-S-LPS of *S. flexneri*, namely 1b, 2a, 3a, 6, and Y.

Each Ac_3_-S-LPS is a macromolecule, and at the same time is a full-fledged bioconjugate vaccine. In this respect, the O-PS conjugate is naturally produced by the bacterial cell and contains a built-in adjuvant-lipid A domain, the structure of which is deacylated to an apyrogenic form. The low content of pyrogens in PLVF preparation was confirmed using the rabbit pyrogenicity test. Preclinical toxicology studies in rabbits demonstrated the complete absence of signs of acute and chronic toxicity, local irritant action, damage to internal organs, and changes in biochemical and hematological parameters when PLVF was administered at a dose of 125 μg in 0.5 mL intended for human administration.

Thus, we achieved a significant extension of the dose range for a single parenteral administration to animals of Ac_3_-S-LPS as an active substance of PLVF to over 100 μg. Under the chronic toxicity regimen, multiple administration up to 875 μg of Ac_3_-S-LPS did not cause any endotoxic effects. These data once again demonstrate the critical difference between Ac_3_-S-LPS and natural endotoxin, the safe parenteral dose of which is sharply lower—just a few nanograms.

We used SDS-PAGE and mass spectrometry to determine the structure of Ac_3_-S-LPS of *S. flexneri* 1b, 2a, 3a, 6, and Y. Ac_3_-S-LPS from PLVF contains the long-chain S-LPS (O-PS chain length is about 20 repeating units) and has a triacylated lipid A. It should be emphasized that such vaccine grade Ac_3_-S-LPS is completely free from the most endotoxic highly acylated molecules of LPS—its hexa- and penta-acylated derivatives. The structure of the polysaccharide component of Ac_3_-S-LPS was determined using ^13^C NMR and was distinct for each serotype.

Each Ac_3_-S-LPS of PLVF was identified using a homologous monovalent antiserum. The serological properties of PLVF components and nLPS of the same *S. flexneri* serotypes were similar. Thus, the methods used to obtain Ac_3_-S-LPS did not cause the alteration of antigenic determinants.

In double immunized mice, we have registered high IgG levels (16-28-fold rise) to all components of PLVF without an aluminum hydroxide adjuvant. High immune response in mice correlated with protection against dysentery eye infection in guinea pigs. We have registered 50–75% efficacy against dysenteric keratoconjunctivitis after double immunization of guinea pigs. In a recent paper [21], a 75% protection rate in the *S. flexneri 2a* Sereny test after triple subcutaneous immunization with Invaplex _AR-Detox_ vaccine was reported. Thus, in addition to the 2a component, the simultaneous induction of a systemic immune response in mice and mucosal protection in guinea pigs has now been confirmed for four more *S. flexneri* O-antigens.

The correctness of the choice of Ac_3_-S-LPS with long-chain OPS as a protective antigen of PLVF was confirmed by the presence in the sera of mice immunized with the vaccine of significant amounts of bactericidal antibodies to each of the homologous virulent strains of *S. flexneri*, serotypes 1b, 2a, 3a, 6, and Y.

The results of this preclinical study substantiated a wide dose interval for the safe administration of the candidate vaccine product and, thus, successfully validated the design of a pentavalent vaccine combination. If the goal of protection in the field against Flexner’s shigellosis using Ac_3_-S-LPS vaccine is achieved, the problem of comprehensive protection against shigellosis may be solved by the combined use of PLVF and the first available vaccine against Sonne’s shigellosis. This vaccine is based on *S. sonnei* exopolysaccharide, which elicits protective antibodies, and has been successfully used for prophylactic immunization for 15 years. In addition, the comprehensive data on the safety of PLVF offers the possibility of extending our vaccine construction strategy to various serotypes of Enterobacteriaceae.

## Figures and Tables

**Figure 1 vaccines-11-00345-f001:**
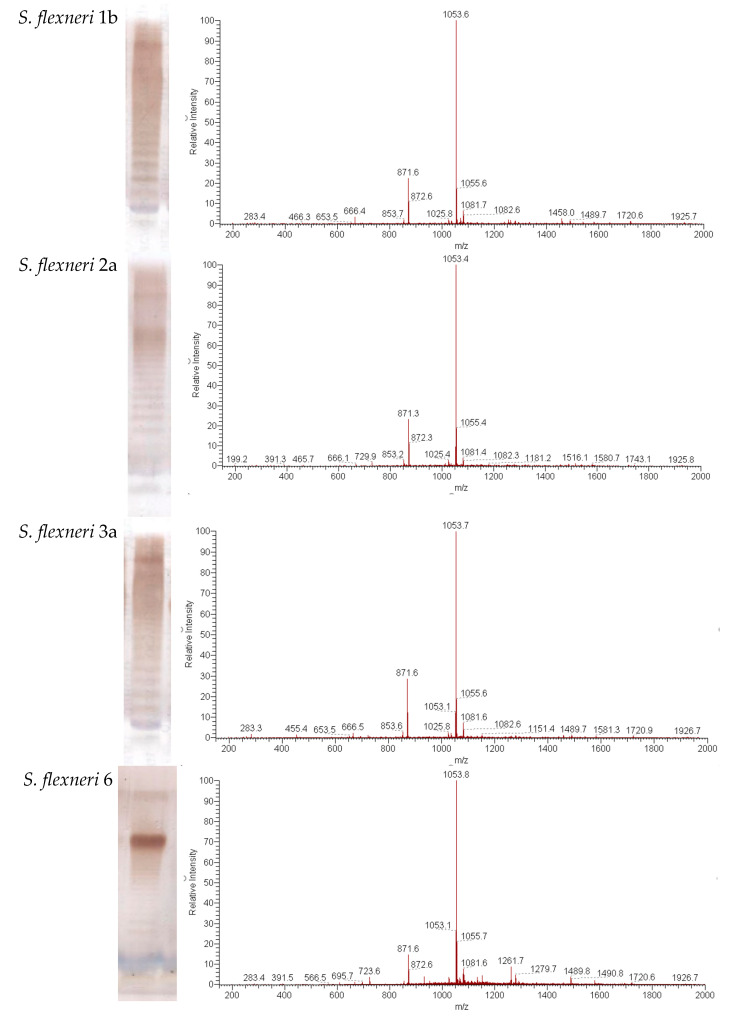
Structural analysis of *S. flexneri* Ac_3_-S-LPS isolated from serotypes 1b, 2a, 3a, 6, and Y. (**a**) Silver-stained SDS-PAGE was used to determine the length of the O-PS chain and (**b**) negative ion mass spectrometry was used to determine the amount of fatty acids. Lipid A samples were released from the modified S-LPS of *S. flexneri* by mild acid hydrolysis (2% AcOH, 100 °C, 1 h).

**Figure 2 vaccines-11-00345-f002:**
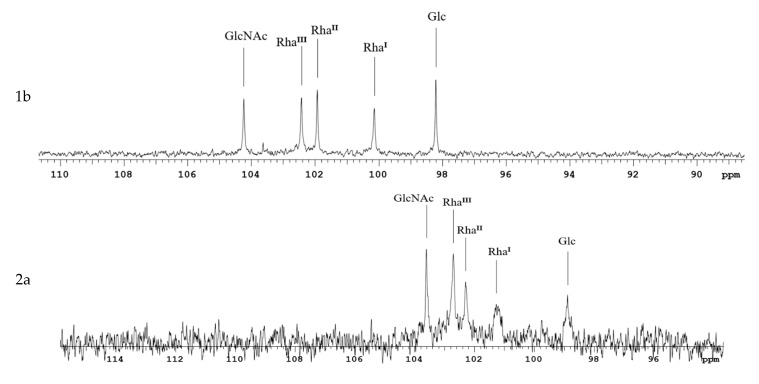
Position of the characteristic signals in the anomeric carbon region of the ^13^C NMR spectra of *S. flexneri* Ac_3_-S-LPS isolated from serotypes 1b, 2a, 3a, 6, and Y. The region for CO resonances is not shown. The assignment of rhamnose signals corresponds to their numbers in the structures (Appendix A).

**Figure 3 vaccines-11-00345-f003:**
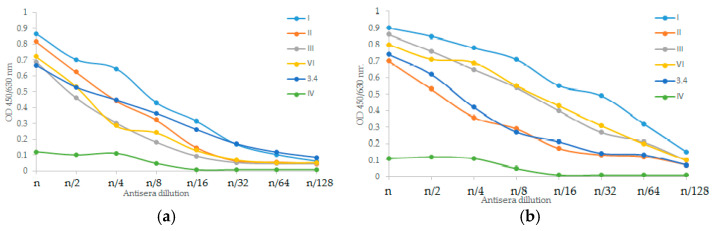
Immunochemical characterization of PLVF and nLPS of *S. flexneri*. Specific activity was investigated by ELISA of: (**a**) PLVF and (**b**) nLPS of *S. flexneri* 1b, 2a, 3a, 6, and Y using monovalent antisera against type-specific antigens I, II, III, and VI, and group-specific antigen 3,4, respectively. Type IV-specific antiserum was used as a negative control. (n − starting serum dilution).

**Figure 4 vaccines-11-00345-f004:**
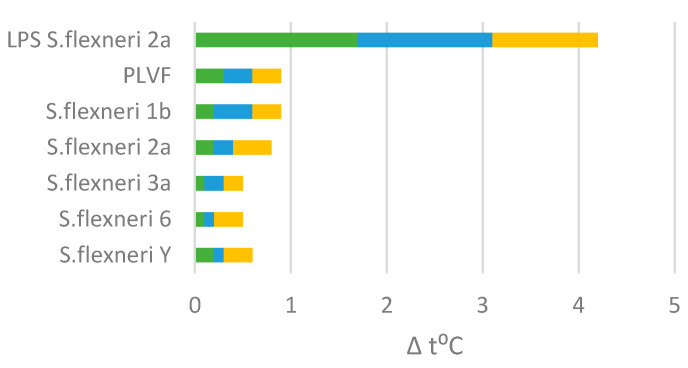
Comparative study of febrile response to PLVF, each Ac_3_-S-LPS vaccine component, and nLPS of *S. flexneri* 2a in the rabbit pyrogen test. For each group, the cumulative temperature rise was calculated using three rabbits. The stacked color bars represent each individual rabbit febrile response within a group.

**Figure 5 vaccines-11-00345-f005:**
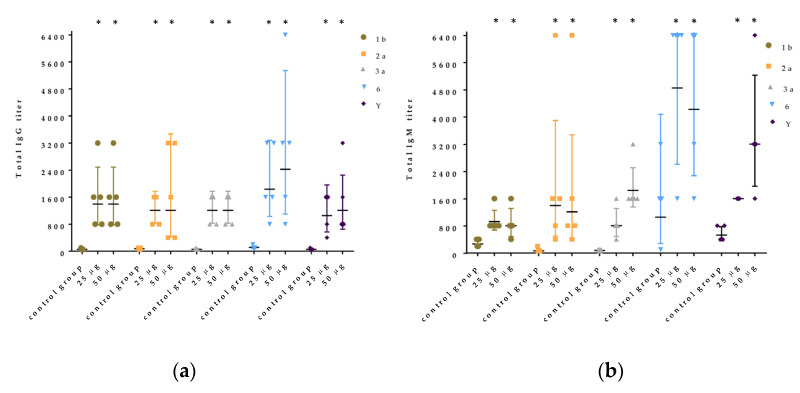
Immunogenicity in mice of individual components of PLVF—five Ac_3_-S-LPSs from *S. flexneri* 1b, 2a, 3a, Y, and 6. (**a**) Total IgG responses after vaccination with a dose of 25 or 50 μg, (**b**) IgM responses after vaccination with a dose of 25 or 50 μg. Results shown as mean ± SD. Significant difference between control and experimental group values is indicated (* *p* < 0.05).

**Figure 6 vaccines-11-00345-f006:**
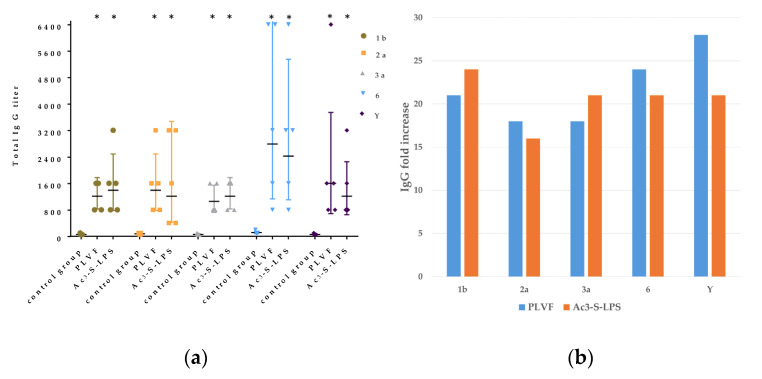
Comparative analysis of serotype-specific total IgG responses against PLVF at a dose of 125 μg per mouse and its individual Ac_3_-S-LPS components at a dose of 25 μg per mouse (Ac_3_-S-LPS 1b, 2a, 3a, 6, and Y). The immunogenicity of each Ac_3_-S-LPS was tested using wells coated with the corresponding antigen. (**a**) Total IgG titers in mice after immunization with PLVF or its individual components, (**b**) increase in specific IgG titers relative to unimmunized control group. (* *p* ≤ 0.05 compared with unimmunized control).

**Figure 7 vaccines-11-00345-f007:**
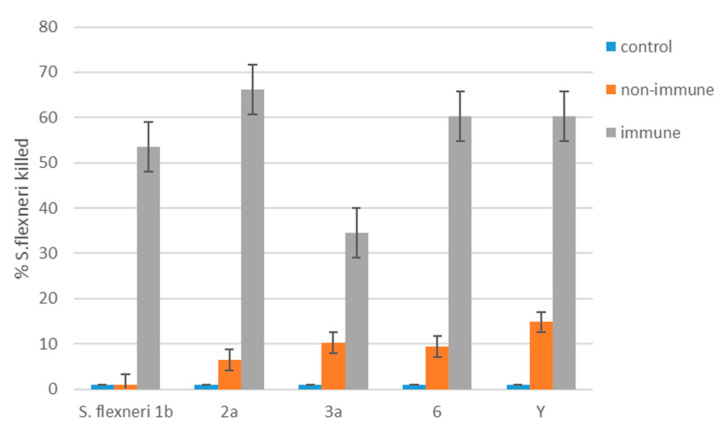
Serum bactericidal antibody assay on immune sera from mice immunized with PLVF at a dose of 125 μg against virulent *S. flexneri* strains, serotypes 1b, 2a, 3a, 6, and Y. Wells containing *S. flexneri* bacteria with guinea pig complement with no serum were used as complement control. Non-immune wells contained sera from naïve mice. Data represent mean SBA rate for each serotype ± SEM.

**Figure 8 vaccines-11-00345-f008:**
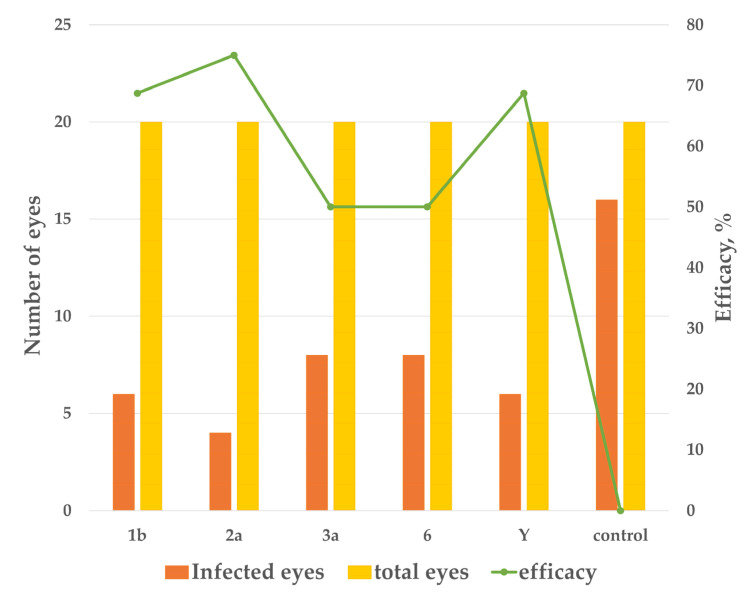
Protective immunity against separate infection with each of *S. flexneri* 1b, 2a, 3a, 6, and Y serotype was induced in guinea pigs by systemic immunization with PLVF. The control group of guinea pigs was injected with saline.

## Data Availability

The data presented in this study are available on request from the corresponding author.

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
