# Peer review of "A Pentavalent Shigella flexneri LPS-Based Vaccine Candidate Is Safe and Immunogenic in Animal Models"

_vaccines, 2023, doi:10.3390/vaccines11020345_

Round 1
Reviewer 1 Report
I think it is difficult to develop a pentavalent LPS candidate vaccine against S. flexneri 1b, 2a, 3a, 6, Y. I would like to provide our suggestion for your reference.
1、The authors should clarify why they choose S. flexneri 1b, 2a, 3a, and 6 y why not choose S. flexneri 1b, 2a, 2b, 3a, and 6. We know that sero-types/serogroups isolated from approximately 90% of patients with dysentery include S. flexneri 1b, 2a, 2b, 3a, and 6 in line 59-60.
2、In the guinea pig keratoconjunctivitis model, the level of protection was 60% to 80% against challenge with all five S. flexneri serotypes. In fact, the control group of guinea pigs injected with saline solution which provide 20% protection. So, I think the protection rates is not correct.
3、In the rabbit pyrogen test, the authors should provide specific information about the rabbits numbers and why choose 0.025ug/kg dose because they use 125 μg per mouse of PLVF.
4、In the Toxicology study and Pyrogenicity test, both the rabbits have the same weight with 2708 ± 71 g. Please explain the data.
5、The authors provided antibodies titers of 15 Days after the secondary immunization in mice of the PLVF. I think the time is too short to value the duration for the PLVF immunogenicity study.
6、In efficacy evaluation of PLVF, vaccine was immunized twice at an interval of 10 days. In immunogenicity in mice, vaccine was immunized twice at an interval of two weeks. Why the authors design the study at different time interval.
7、The authors have not provide why choose 125 μg as vaccine dose, whether they observe the results of different doses in different time in different routes.
8、The authors should explain whether the guinea pig keratoconjunctivitis model can be used as a gold standard model for the efficacy study.
Author Response
Reviewer #1:
I think it is difficult to develop a pentavalent LPS candidate vaccine against S. flexneri 1b, 2a, 3a, 6, Y. I would like to provide our suggestion for your reference.
We thank the reviewer for careful review of our manuscript and appreciate his comments and suggestions.
- The authors should clarify why they choose S. flexneri 1b, 2a, 3a, and 6 y why not choose S. flexneri 1b, 2a, 2b, 3a, and 6. We know that sero-types/serogroups isolated from approximately 90% of patients with dysentery include S. flexneri 1b, 2a, 2b, 3a, and 6 in line 59-60.
We thank the reviewer for raising this issue. There is a consensus in the field based on the analysis of Shigella O antigens and cross protection studies that inclusion of S. flexneri 2a, 3a, and 6 in the vaccine will provide cross protection against the other 11 S. flexneri serotypes because of shared group antigens [Noriega et al. 1999; Levine et al. 2007]. This concept was applied to the proposed combinations of 4- or 5-valent bioconjugate vaccines, GMMA, Invaplex, in which two components belonging to the same serotype are not included (for example, 2a and 2b). When we designed our pentavalent S. flexneri vaccine candidate we also used the same approach.
- In the guinea pig keratoconjunctivitis model, the level of protection was 60% to 80% against challenge with all five S. flexneri serotypes. In fact, the control group of guinea pigs injected with saline solution which provide 20% protection. So, I think the protection rates is not correct.
We thank the reviewer for pointing this out. We added the following phrase in the Results section: “In unimmunized control animals, at least 80% of the eyes were infected depending on the serotype of the virulent strain of S. flexneri.” Lines 367-368.
We corrected the efficacy of vaccine preparations taking in account the ID80 dose. The efficacy of PLVF was calculated using the formula: PLVF Efficacy = 100 × (Control attack rate – PLVF attack rate) / Control attack rate, attack rate= number of infected eyes/ total eyes. The efficacy of PLVF was 69, 75, 50, 50, and 69% against S. flexneri serotypes 1b, 2a, 3a, 6, Y, respectively.
We changed efficacy values in the text accordingly: on line 26 “the efficacy was 50% to 75%”, lines 348-349 “the efficacy was 69%, 75%, 50%, 50%, and 69% against S. flexneri serotypes 1b, 2a, 3a, 6, Y”, and line 450 “registered 50-75% efficacy”.
- In the rabbit pyrogen test, the authors should provide specific information about the rabbits numbers and why choose 0.025 ug/kg dose because they use 125 μg per mouse of PLVF.
While we understand the reviewer’s concern, we would like to clarify this issue. We updated information on the number of rabbits which temperature data are presented in Figure 4. The following sentence was added to the Materials and Methods. “Rabbits were randomized by weight into 7 groups. Three rabbits per group were intravenously injected with PLVF, each Ac3-S-LPS vaccine component or unmodified nLPS of S. flexneri 2a (as pyrogenicity control) at a dose of 0.025 µg kg-1.” Lines 157-160.
Regarding the dose of 0.025 ug/kg. It was included into the pharmacopoeial parameters for the control of capsular polysaccharide vaccines containing highly endotoxic native LPS. This relates to Vi polysaccharide-containing typhoid vaccine and meningococcal A vaccine. The fact that modified LPS is non-pyrogenic and was tested at the same dose as capsular polysaccharide indicates its safety.
- In the Toxicology study and Pyrogenicity test, both the rabbits have the same weight with 2708 ± 71 g. Please explain the data.
We thank the reviewer for pointing out our mistake. This was a copy-paste error. We corrected the text accordingly. “RPT was conducted on Chinchilla rabbits (aged 3 months at the start of experiment and weighing 2.8-3.05 g) in accordance with European Pharmacopoeia requirements [16].” Lines 156-157.
- The authors provided antibodies titers of 15 Days after the secondary immunization in mice of the PLVF. I think the time is too short to value the duration for the PLVF immunogenicity study.
The carbohydrate-specific immune response is characterized by an earlier development compared to the response against protein antigens. Antibody producing cells appear on the 3rd-4th day after immunization, and specific IgG and IgM can be already determined on the 7th day.
- In efficacy evaluation of PLVF, vaccine was immunized twice at an interval of 10 days. In immunogenicity in mice, vaccine was immunized twice at an interval of two weeks. Why the authors design the study at different time interval.
In guinea pigs, we investigated the mucosal immune response on the conjunctival mucosa with the route of vaccine administration subcutaneous dorsal area. Systemic immune response was studied in mice, with intraperitoneal route of vaccine administration. The protocol, number of doses and timing were worked out in advance in order to obtain the protective effect of the vaccine preparation in guinea pigs, and production of specific antibodies in the mouse sera.
- The authors have not provide why choose 125 μg as vaccine dose, whether they observe the results of different doses in different time in different routes.
Numerous studies of oral Shigella vaccines have unfortunately proved to be unpromising. Today the main trend is parenteral administration of Shigella vaccines. In the discussion we explained that we are trying to maximize the dose of subcutaneous administration of the drug to a human subject, while maintaining a high safety profile of the administered dose.
For good serological coverage, the maximum possible number of serotypes, five, was included in the preparation. The immunogenicity of a dose of 25 µg of Ac-3 LPS in humans has been previously confirmed. The complete safety of the composition 5x25=125 µg was confirmed in this study in various ways. Therefore, the expected high immunogenicity for dose 125 µg was confirmed for five serotype-specific responses. It can be administered to a person as providing the maximum activation of adaptive immunity. The use of lower doses at this stage is not very promising and problematic. From our point of view, studies on dose reduction can be carried out if we achieve a level of immune activation adequate for field protection.
- The authors should explain whether the guinea pig keratoconjunctivitis model can be used as a gold standard model for the efficacy study.
“The guinea pig keratoconjunctivitis model, the basis for the Sereny test, remains the most reliable in vivo indicator of virulence of Shigella strains and immunogenicity and protective efficacy of Shigella vaccine candidates.”(Hanson et al., 2001). This statement remains to be true in spite the fact that some surrogate mouse models have been developed more recently. Thus, we use the guinea pig keratoconjunctivitis model for pre-clinical studies of protective efficacy of our vaccine formulations.
Reviewer 2 Report
Overall this is a well-written and interesting study. The English tends to be less smooth in the discussion.
1) In Materials and Methods, Pyrogenicity, it would be helpful to mention how many rabbits were used for the pyrogenicity test and how it was done. Right now this information is in the Figure 4 caption.
2) In Materials and Methods, Efficacy evaluation of PLVF, There is a sentence, "Each group was immunized with one of S. flexneri serotypes" If this refers to placing S. flexneri in the eye, "immunized" is incorrect, and should be changed to "inoculated."
Also, the calculation of the protection rate should not be the percentage of non-infected eyes per group, as the control group had only 80% infected eyes. The suggested formula for vaccine efficacy / protection for the guinea pig study is:
(Risk among unvaccinated group − risk among vaccinated group) / (divided by) Risk among unvaccinated group
The results will show a lower protection rate than currently stated, which should also be corrected in the discussion.
3) In Materials and Methods, Immunogenicity in mice and SBA, there were 75 mice total and 5 per group. 1 group for PLVF and 2 groups for each serotype (2 different doses). That is 11 groups. What were the other 4 groups? Were there control mice? Were control mice given PBS injection or not injected at all? Please clarify the presence of control groups and account for all 15 group assignments.
In the SBA description, it is said, "Negative control wells included bacteria with guinea pig complement." "Bacteria" is vague, and if the wells were also S. flexneri of each serotype, that should be stated. If the negative control wells were without mouse sera, that should be stated. For example, it could say, Negative control wells included S. flexneri with guinea pig complement and no sera. The same clarification should be used in the caption to Figure 9.
4) In Results, Immunogenicity in mice, please clarify whether the increases in IgG titers were relative to baseline in the same mice or relative to controls.
5) In Results, keratoconjunctivitis, as mentioned previously, the percent protection should be relative to controls, not relative to 20 eyes.
6) Toxicity: It is hard to believe that there were no absolutely no signs of inflammation such as mild lymphocytic infiltration or even just tissue disruption and healing after 7 days of injections. That is just a comment.
7) Figure 5 and Figure 6. The legend colors indicate the IgG responses to different serotypes, but the X axis should be labeled with the separate Ac3-S-LPSs used for vaccination for each group.
8) There are not many references cited about the origin of the methods used to isolate, detoxify and purify the vaccine substance from the cultures and LPS fraction. If there are references, please add them.
9) The English tends to falter in the discussion from line 384 to line 421 and needs editing, whereas the rest of the paper seems very well-written.
10) You could consider discussing how new carbohydrate synthetic methods could build on your work perhaps by improving the purity of the vaccine or isolating the specific epitopes, or whether your direct isolation approach from S. flexneri has specific advantages.
Author Response
Reviewer #2:
Overall this is a well-written and interesting study. The English tends to be less smooth in the discussion.
We would like to thank the reviewer for his/her positive and insightful comments on the manuscript.
- In Materials and Methods, Pyrogenicity, it would be helpful to mention how many rabbits were used for the pyrogenicity test and how it was done. Right now this information is in the Figure 4 caption.
We thank the reviewer for pointing this out. We modified the text accordingly. “RPT was conducted on 21 adult Chinchilla rabbits (aged 3 months at the start of experiment and weighing 2.8-3.05 g) in accordance with European Pharmacopoeia requirements [16]. Rabbits were randomized by weight into 7 groups. Three rabbits per group were intravenously injected with PLVF, each Ac3-S-LPS vaccine component or unmodified nLPS of S. flexneri 2a (as pyrogenicity control) at a dose of 0.025 µg kg-1.” Lines156-160.
- In Materials and Methods, Efficacy evaluation of PLVF, There is a sentence, "Each group was immunized with one of S. flexneri serotypes" If this refers to placing S. flexneri in the eye, "immunized" is incorrect, and should be changed to "inoculated."
We thank the reviewer for pointing this out. We corrected this mistake. ”Each group was inoculated with one of S. flexneri 1b, 2a, 3a, 6, Y serotypes.” Lines 233-234.
- Also, the calculation of the protection rate should not be the percentage of non-infected eyes per group, as the control group had only 80% infected eyes. The suggested formula for vaccine efficacy / protection for the guinea pig study is:
(Risk among unvaccinated group − risk among vaccinated group) / (divided by) Risk among unvaccinated group
We thank the reviewer for this thoughtful comment. We calculated the efficacy of PLVF was using the following formula: PLVF Efficacy = 100 × (Control attack rate – PLVF attack rate) / Control attack rate, attack rate= number of infected eyes/ total eyes
The proposed formula includes an "Attack rate" for a more accurate result. The difference compared to the proposed option is small, the minimum and maximum efficiency are the same level - 50% and 75%, respectively.
- The results will show a lower protection rate than currently stated, which should also be corrected in the discussion.
We changed efficacy values in the text accordingly: on line 26 “the efficacy was 50% to 75%”, lines 348-349 “the efficacy was 69%, 75%, 50%, 50%, and 69% against S. flexneri serotypes 1b, 2a, 3a, 6, Y”, and line 450 “registered 50-75% efficacy”.
- In Materials and Methods, Immunogenicity in mice and SBA, there were 75 mice total and 5 per group. 1 group for PLVF and 2 groups for each serotype (2 different doses). That is 11 groups. What were the other 4 groups? Were there control mice? Were control mice given PBS injection or not injected at all? Please clarify the presence of control groups and account for all 15 group assignments.
We thank the reviewer for pointing this out. Indeed, the work presented data obtained from 12 groups, 5 mice each (10 groups were inoculated with two doses of 25 μg and 50 μg of Ac3-S-LPS 1b, 2a, 3a, 6, Y; and one group per PLVF and control). Control animals were given saline. 60 mice in total.
We corrected our mistake. ”Sixty (СВА x С57ВL/6) F1 female mice (5 mice per group, 8-week-old at the start of experiment and weighing 18 ± 0.3 g) were purchased from the Federal State Budgetary Institution of Science "Scientific Center for Biomedical Technologies of the Federal Medical and Biological Agency", Russia and immunized intraperitoneally with 125 μg per mouse of PLVF or with two doses of 25 μg and 50 μg of Ac3-S-LPS 1b, 2a, 3a, 6, Y.” Lines 199-203.
- In the SBA description, it is said, "Negative control wells included bacteria with guinea pig complement." "Bacteria" is vague, and if the wells were also S. flexneri of each serotype, that should be stated. If the negative control wells were without mouse sera, that should be stated. For example, it could say, Negative control wells included S. flexneri with guinea pig complement and no sera. The same clarification should be used in the caption to Figure 9.
We thank the reviewer for this suggestion. We modified text accordingly. “Assay included a complement control wells containing S. flexneri bacteria with guinea pig complement with no serum. This complement control was used to define 0 % killing in the SBA killing rate calculation.” Lines 216-219.
- In Results, Immunogenicity in mice, please clarify whether the increases in IgG titers were relative to baseline in the same mice or relative to controls.
We thank the reviewer for pointing this out. Indeed, we calculated the increases in IgG titers relative to control. We corrected Figure 6 accordingly. “(b) Increase in specific IgG titers relative to unimmunized control group.” Lines 334-335.
- In Results, keratoconjunctivitis, as mentioned previously, the percent protection should be relative to controls, not relative to 20 eyes.
We changed efficacy values in the text accordingly on lines 348-349 “the efficacy was 69%, 75%, 50%, 50%, and 69% against S. flexneri serotypes 1b, 2a, 3a, 6, Y.
- Toxicity: It is hard to believe that there were no absolutely no signs of inflammation such as mild lymphocytic infiltration or even just tissue disruption and healing after 7 days of injections. That is just a comment.
We understand the reviewer’s concern. Histology of the injection site was done 7 days after injection. It is likely that slight changes in tissues already disappeared and were not observed by a histopathologist. However, visual inspection showed no signs of local inflammation either.
- Figure 5 and Figure 6. The legend colors indicate the IgG responses to different serotypes, but the X axis should be labeled with the separate Ac3-S-LPSs used for vaccination for each group.
Unfortunately, the software (Prizm) does not allow to specify the individual compound name under each column. Therefore, we tried wherever possible to indicate that the immune response was against the antigen with which animals were vaccinated. We assume that the color pattern of the chart allows to identify each Ac3-LPS serotype.
- There are not many references cited about the origin of the methods used to isolate, detoxify and purify the vaccine substance from the cultures and LPS fraction. If there are references, please add them.
We added an additional reference.
- Aparin, P.G.; Lvov, V.L.; Elkina, S.I.; Golovina, M.E. Modified lipopolysaccharide of endotoxic bacteria (options), a combination of modified lipopolysaccharides (options) and a vaccine (options) including them and a pharmaceutical composition (options). WO2014196887A1 2014.
- The English tends to falter in the discussion from line 384 to line 421 and needs editing, whereas the rest of the paper seems very well-written.
We thank the reviewer for pointing this out. We extensively edited the Discussion section.
- You could consider discussing how new carbohydrate synthetic methods could build on your work perhaps by improving the purity of the vaccine or isolating the specific epitopes, or whether your direct isolation approach from S. flexneri has specific advantages.
Improvements in quality and purity of polysaccharide antigens for vaccination can be achieved several ways. Our direct isolation approach, bioconjugation and synthetic methods are the most promising. Synthetic methods definitely produce the purest preparations. However, design of synthetic carbohydrates has to account for all structural peculiarities that determine their natural analog's antigenicity. This task is especially complex for S. flexneri due to existence of multiple serotypes. Currently, no synthetic carbohydrates have been made for S. flexneri, so it is too early to discuss their clear advantages or disadvantages. It can be assumed that the weak points of synthetic vaccines may be the cost of production and the yield of the final product.
Recombinant construction of glycoconjugates in bacteria is very promising approach in development of modern vaccines against bacterial enteric infections. Synthesis of bioconjugates became more and more effective, scale affordable, money-saving. But well-known for carbohydrates and LPS problem of polydispersity also exist for bioconjugates, which may differ in structural characteristics from batch to batch.
LPS is naturally synthesized bioconjugate of O-PS antigen and lipid A adjuvant domain. We undertook several attempts to obtain more homogenous LPS, for example started to work only with S-form. Homogenous form of bioconjugate will be giving more reliable immune response. May be this approach will be useful for processing recombinant bioconjugates.
Reviewer 3 Report
The submitted manuscript aims to evaluate the safety and immunogenicity of a pentavalent Shigella flexneri LPS-based vaccine. The authors conclude that the vaccine is nontoxic and immunogenic at multiple doses. Additionally, protective efficacy is demonstrated in vivo.
The manuscript addresses a timely topic which fits within the scope of Vaccines. The manuscript was very well organized, and the data was presented in a cohesive and clear manner. The authors utilized appropriate models for evaluation of various vaccine effects. Overall, this is an intriguing and well-presented manuscript. Minor comments are as follows:
Minor Comments:
Title: I suggest making the title more direct in relation to the findings, e.g. “A multivalent LPS-based, multivalent Shigella flexneri vaccine candidate is safe and immunogenic”
--Lines 68-69: I suggest adding appropriate citations to bolster the statement that “protective immunity to Shigella is induced by O-polysaccharide…”
-Line 76: I suggest changing “rises” to levels or titers for clarity.
-Line 76: I feel that the wording of this sentence (e.g. stating safety assessment and safety profile back to back) is redundant.
-Materials and Methods: I suggest adding more detail regarding animal husbandry and housing conditions to comply with ARRIVE guidelines and for transparency.
-Materials and Methods: I think it would be helpful for the reader if more details were given about the histological methods.
-Figure 5: The statistical asterisk indicators are not formatted correctly.
-Funding: Perhaps the authors could state their funding source.
Author Response
Reviewer #3:
The submitted manuscript aims to evaluate the safety and immunogenicity of a pentavalent Shigella flexneri LPS-based vaccine. The authors conclude that the vaccine is nontoxic and immunogenic at multiple doses. Additionally, protective efficacy is demonstrated in vivo.
The manuscript addresses a timely topic which fits within the scope of Vaccines. The manuscript was very well organized, and the data was presented in a cohesive and clear manner. The authors utilized appropriate models for evaluation of various vaccine effects. Overall, this is an intriguing and well-presented manuscript.
We thank the reviewer for careful review of our manuscript and appreciate his comments and suggestions.
Minor comments are as follows:
- Title: I suggest making the title more direct in relation to the findings, e.g. “A multivalent LPS-based, multivalent Shigella flexneri vaccine candidate is safe and immunogenic”
We modified the title according to the reviewer’s suggestion. ”A pentavalent Shigella flexneri LPS-based vaccine candidate is safe and immunogenic in animal models.”
- Lines 68-69: I suggest adding appropriate citations to bolster the statement that “protective immunity to Shigella is induced by O-polysaccharide…”
We added an additional reference.
- Robbins, J.B.; Chu, C.; Schneerson, R. Hypothesis for vaccine development: Protective immunity to enteric diseases caused by nontyphoidal salmonellae and shigellae may be conferred by serum igg antibodies to the o-specific polysaccharide of their lipopolysaccharides. Clin. Infect. Dis. 1992, 15, 346–361, doi:10.1093/clinids/15.2.346.
- Line 76: I suggest changing “rises” to levels or titers for clarity.
We thank the reviewer for pointing this out. We corrected text accordingly. “Antibody levels after immunization of mice with PLVF were assessed.” Line 76.
- Line 76: I feel that the wording of this sentence (e.g. stating safety assessment and safety profile back to back) is redundant.
We thank the reviewer for pointing this out. We corrected text accordingly. “Preclinical safety profile of PLVF was studied in rabbits.” Lines 75-76.
- Materials and Methods: I suggest adding more detail regarding animal husbandry and housing conditions to comply with ARRIVE guidelines and for transparency.
We added an additional information in the Materials and Methods on lines 145-152.
“The study on animal models was carried out in accordance with the ethical principles approved by the order of the Ministry of Health of the Russian Federation No. 199n from 04 January 2016. The study protocols were approved by the local ethics committees of the research organizations. Animals were housed in accredited animal facilities with free access to food and water. Before the start of a study, animals were placed in a separate room for a period of quarantine (14-21 days, depending on the animal species) and health-monitored.”
- Materials and Methods: I think it would be helpful for the reader if more details were given about the histological methods.
We added an additional information in the Materials and Methods on lines 183-187.
“Tissue samples for histological examination were fixed in 10% neutral buffered formalin, dehydrated in ascending concentration of alcohol, and embedded in paraffin. Paraffin sections 5 μm thick were cut on a SM 2000R microtome (Leica, Germany), stained with hematoxylin and eosin, and examined using a DM1000 microscope (Leica, Germany).”
- Figure 5: The statistical asterisk indicators are not formatted correctly.
We thank the reviewer for pointing this out. We corrected formatting accordingly.
- Funding: Perhaps the authors could state their funding source.
We added an additional information about funding source. Lines 483-486.
“This study was in part funded by a government grant No. 14.N08.11.0009 for carrying out research and development, based on the decision of the Competition Commission of the Ministry of Education and Science of the Russian Federation No. 2013-14-241/01 (protocol dated July 26, 2013 No. 0173100003713000245-3-03).”
Round 2
Reviewer 1 Report
I have no further suggestions for the present manuscript.